# Chemical Genetic Screen in *Drosophila* Germline Uncovers Small Molecule Drugs That Sensitize Stem Cells to Insult-Induced Apoptosis

**DOI:** 10.3390/cells10102771

**Published:** 2021-10-16

**Authors:** Julien Roy Ishibashi, Riya Keshri, Tommy Henry Taslim, Daniel Kennedy Brewer, Tung Ching Chan, Scott Lyons, Anika Marie McManamen, Ashley Chen, Debra Del Castillo, Hannele Ruohola-Baker

**Affiliations:** 1Department of Biochemistry, University of Washington, Seattle, WA 98195, USA; jishi@uw.edu (J.R.I.); riyakeshri@iisc.ac.in (R.K.); T.TASLIM1234@edmail.edcc.edu (T.H.T.); daniel.kbrewer@yahoo.com (D.K.B.); tching99@uw.edu (T.C.C.); snlyons@earthlink.net (S.L.); mcanika@uw.edu (A.M.M.); ashley94c@yahoo.com (A.C.); debradel@uw.edu (D.D.C.); 2Institute for Stem Cell and Regenerative Medicine, School of Medicine, University of Washington, Seattle, WA 98109, USA

**Keywords:** *Drosophila*, germline, stem cells, apoptosis, cancer, quiescence, small molecule, radiation

## Abstract

Cancer stem cells, in contrast to their more differentiated daughter cells, can endure genotoxic insults, escape apoptosis, and cause tumor recurrence. Understanding how normal adult stem cells survive and go to quiescence may help identify druggable pathways that cancer stem cells have co-opted. In this study, we utilize a genetically tractable model for stem cell survival in the *Drosophila* gonad to screen drug candidates and probe chemical-genetic interactions. Our study employs three levels of small molecule screening: (1) a medium-throughput primary screen in male germline stem cells (GSCs), (2) a secondary screen with irradiation and protein-constrained food in female GSCs, and (3) a tertiary screen in breast cancer organoids in vitro. Herein, we uncover a series of small molecule drug candidates that may sensitize cancer stem cells to apoptosis. Further, we have assessed these small molecules for chemical-genetic interactions in the germline and identified the NF-κB pathway as an essential and druggable pathway in GSC quiescence and viability. Our study demonstrates the power of the *Drosophila* stem cell niche as a model system for targeted drug discovery.

## 1. Introduction

In adult organisms, stem cells maintain organ homeostasis by differentiating into diverse specialized cells and through the property of self-renewal. In contrast, tumor-initiating cells, also known as cancer stem cells (CSCs), contribute to tumor homeostasis by producing heterogeneous cancer cells in a solid tumor. Both adult stem cells and CSCs show resistance to chemical- or radiation-induced apoptosis and can enter “quiescence”, a reversible state of proliferative arrest. Quiescence of adult stem cells during cancer treatment accounts for most of the adverse side effects, whereas the reversibility of CSC quiescence increases oncogenicity and lethality [1]. Hence, there’s an emerging need to develop targeted therapies that eradicate CSCs by preventing quiescence and/or promoting apoptosis. Several pharmacological drugs such as salinomycin (affects proliferation), LGK-974 (affects Wnt pathway), and MK-0752 (affects Notch pathway) are under clinical trial for ablation of the CSCs [2,3,4]. However, due to their transient and diverse nature, studying CSCs’ response to drug treatment in vivo remains a challenge. Decrypting normal adult stem cells survival and quiescence after insult may help identify druggable pathways utilized also by cancer stem cells.

Signaling pathways such as mTOR, NF-κB, Notch have been found to promote stemness in both healthy adult stem cells and cancer stem cells. mTOR/PI3K/Akt signaling has previously been characterized as a crucial regulator of a well established adult stem cell model, *Drosophila* germline stem cell (GSC) maintenance and quiescence [5,6,7,8]. mTOR signaling has similarly been reported as a master regulator for maintaining CSCs in prostate and breast cancers [9,10]. Several studies have also reported that, in various cancers such as glioblastoma, acute myelogenous leukemia and prostate cancers CSCs, both canonical and non-canonical NF-κB signaling contribute to tumor growth and metastasis [11,12,13]. Furthermore, in certain contexts, NF-κB transcriptional activity protects against apoptosis [14,15,16,17], while inhibition of NF-κB with SN50, or inhibition of mTOR with rapamycin, has been shown to reduce the sphere forming ability of the glioblastoma CSCs and pancreatic CSCs, respectively [18,19]. Though NF-κB/Toll signaling is thought to function in female GSC aging [20], the context-dependent role of NF-κB signaling in GSC quiescence and survival remains unclear. Therefore, targeting these signaling pathways is critical for future therapeutic development.

We have previously shown that Drosophila GSC can enter protective insult-induced quiescence, similar to that seen in cancer. Therefore, using *Drosophila* as a model organism, we have attempted to screen for small molecule drugs and druggable pathways that could plausibly be targeted to eliminate CSCs. As an example of an easily identifiable stem cell niche, *Drosophila* GSCs have been employed as a genetically tractable model of adult stem cell quiescence and apoptosis in vivo [5,21,22]. Moreover, the size and scalability of fruit fly culture makes them an advantageous model for medium-to-high throughput screens. In our study utilizing *Drosophila* male GSCs, we first screened and identified small molecules with drug-like qualities that bias stem cells towards apoptosis. This is significant, given that male GSCs have been shown to be particularly resistant to chemical-induced apoptosis [21]. Next, we employed female GSCs to further screen and characterize these small molecules as radiosensitizers that potentiate apoptosis following gamma irradiation. The female GSC niche makes a particularly compelling cancer model because apoptotic daughter cells have been found to protect nearby stem cells from apoptosis following insult [21]. Further, we used MCF7 breast cancer cell line, which is known to have a subpopulation of CSCs [23,24], to validate promising small molecules from the *Drosophila* screen in a breast cancer organoid model. We find certain candidate small molecule drugs that potently impair cancer organoid formation. Furthermore, using tissue-specific gene knockdown of NF-κB effector, IKKε, our study identifies NF-κB pathway as an essential and druggable pathway for insult-induced stem cell quiescence. Our work suggests that these small molecule drug candidates impair stem cell viability and promote apoptosis, urging future study of these compounds and their derivatives as potential chemotherapy drugs.

## 2. Materials and Methods

### 2.1. Fly Stocks and Culture Conditions

Flies were cultured at 25 °C on a cornmeal-yeast-agar-medium supplemented with wet yeast [5,21]. The following stocks were obtained from the Bloomington Drosophila Stock Center at Indiana University: w^1118^ (RRID:BDSC_3605), UAS-Dcr2, w^1118^; nos-Gal4 (RRID:BDSC_25751), w*; UAS-Tsc1 RNAi #1 (RRID:BDSC_35144), UAS-Tsc1 RNAi #2 (RRID:BDSC_54034), UAS-Atg3 RNAi (RRID:BDSC_34359), UAS-IKKεRNAi (RRID:BDSC_34709), UAS-Nhe3 RNAi (RRID:BDSC_60137), and UAS-Ogg1 RNAi (RRID:BDSC_51852).

### 2.2. Ionizing Radiation Treatment

Prior to exposure to gamma-irradiation, 2–4 days old flies (15–18 females: 5–6 males) were kept on cornmeal-yeast-agar-medium augmented with wet yeast for 48 h at 25 °C. On the day of irradiation, 2/3rd of the females and all males were transferred to empty plastic vials and treated with 50 Gy of gamma-irradiation. A Cs-137 Mark I Irradiator was used to administer the proper irradiation dosage, according to instructed dosage chart. The remaining 1/3rd of the females were not irradiated and were dissected within 1 h of the irradiation treatment. After irradiation, the flies were flipped onto a new vial of Standard Diet augmented with wet yeast at 25 °C. Half of the remaining females were dissected at 1 day post-insult (1 dpi). The remaining females were dissected at 2 days post-insult (2 dpi).

### 2.3. High-Throughput Screen of Small Molecules in Low-Melt Agarose Fly Food

The NCI Diversity Set IV was kindly supplied by the National Cancer Institute Developmental Therapeutics Program’s Open Compound Repository, NIH. These NCI compounds were shipped frozen in 20 µL/10 mM stock solutions in 96-well plates and stored at –20 °C. Compounds were dissolved in low-melt agarose fly food to a final concentration of 100 μM [25]. In each well, 3 M flies were drug treated for 72 h and dissected for their testes. Samples were fixed and stained against the protein of interest and assayed for GSC apoptosis.

### 2.4. Cell Death Screen in Grape Juice

Hits from the high-throughput screen of the Diversity Set IV were reordered and kindly supplied by the National Cancer Institute Developmental Therapeutics Program’s Open Compound Repository, NIH, in 5 mg drams. Compounds were diluted in DMSO to 10 mM stock solutions and stored at –20 °C. Prior to drug treatment, 2–4 days old flies (15–18 females: 5–6 males) were kept on cornmeal-yeast-agar-medium augmented with wet yeast for 48 h at 25 °C. Drugs were diluted in grape juice to a final concentration of 200 µM and used within 24 h of mixing. In an empty vial, a 3” × 1” filter paper was soaked in 400 µL of drugged grape juice. Then, 15–18 female and 5–6 male flies were transferred into each vial and flipped into a replenished vial daily for two days before being irradiated. After irradiation, flies were flipped onto a replenished vial daily for another two days. Samples were fixed and stained according to our immunofluorescence protocol and assayed for GSC apoptosis.

### 2.5. Mammosphere Formation Assay

Small molecule candidate drugs were given during breast cancer organoid formation [26]. Cells of a human breast cancer cell line, MCF7, were dissociated with trypsin (0.25%) into single-cells and replated at 26,000 cells/well in ultra-low attachment 6-well plates in DMEM/F12 with 1× Glutamax, 1× Penicilin-Streptomycin, 1× B27, 20 ng/mL EGF, and 10 ng/mL bFGF. Small molecule drugs or vehicles (DMSO) were administered at time of plating. After five days undisturbed, the wells were imaged on a Nikon Widefield High-Resolution Microscope (Nikon Instruments Inc., Melville, NY, USA) and spheres were quantified using ImageJ (Version: 2.0.0-rc-69/1.52p).

### 2.6. Small Molecule Organismal Viability Assay

Prior to drug treatment, 2–4 days old flies (15–18 females: 5–6 males) were kept on cornmeal-yeast-agar-medium augmented with wet yeast for 48 h at 25 °C. To test for organismal lethality, small molecule candidate drugs were diluted in PBS containing 0.5% propionic acid to a final volume/concentration of 1 mL/200 µM or 800 µM directly in an empty vial. Each solution was then mixed with ~570 mg of yeast and smeared up against the wall of the vial once hydrated and nearly homogenous. Then, 15–18 female and 5–6 male flies were placed in each vial and flipped into a replenished vial daily for four days and the number of dead flies was counted each day.

### 2.7. Pathway Analysis in Yeast Paste

Candidate drug stock solutions were reused from the validation of hits in grape juice and reordered as needed from the NIH. Drugs were diluted in PBS containing 0.5% propionic acid to a final volume/concentration of 1 mL/200 µM directly in an empty vial. Each solution was then mixed with ~570 mg of yeast and smeared up against the wall of the vial once hydrated and nearly homogenous. Then, 15–18 females and 5–6 males flies were placed in each vial and flipped into a replenished vial daily for four days.

### 2.8. Immunofluorescence Analysis

Samples were dissected in cold PBS and then fixed in 4% paraformaldehyde for 15 min at room temperature within 30 min of dissection. Samples were then rinsed in PBT (PBS containing 0.2% Triton X-100), and blocked in PBTB (PBT containing 0.2% BSA, 5% normal goat serum) for at least one hour at room temperature. Samples were stored for up to 72 h at 4 °C in PBTB. The following primary antibodies were used: mouse anti-adducin (1:20, RRID:AB_528070, DSHB, Iowa City, IA, USA), mouse anti-Lamin C (1:20, RRID:AB_528339, DSHB, Iowa City, IA, USA), and rabbit anti-cleaved Dcp-1 (1:100, RRID:AB_2721060, Cell Signaling Technology, Danvers, MA, USA). Samples were incubated with primary antibodies for 24 h at 4 °C. After washes with PBT, fluorophore-conjugated secondary antibodies were utilized including anti-rabbit Alexa 488 (1:250, RRID:AB_221544, Thermo Fisher Scientific, Waltham, MA, USA), anti-rabbit Alexa 568 (1:250, RRID:AB_143157, Thermo Fisher Scientific, Waltham, MA, USA), anti-mouse Alexa 488 (1:250, RRID:AB_2534069, Thermo Fisher Scientific, Waltham, MA, USA), anti-mouse Alexa 568 (1:250, RRID:AB_2535773, Thermo Fisher Scientific, Waltham, MA, USA), and anti-rabbit Alexa 647 (1:250, RRID:AB_2535812, Thermo Fisher Scientific, Waltham, MA, USA), for 1.5–2 h at room temperature in the dark. Samples were then incubated with DAPI, diluted with PBT to 2 μg/mL, for 15 min to visualize nuclei, followed by two PBT wash. The samples were mounted in mounting medium (21 mL of Glycerol, 2.4 mL of 10× PBS and 0.468 g of N-Propyl Gallate) and analyzed on a Leica SPE5 confocal and Leica SP8 confocal laser-scanning microscope. Images taken from SPE5 confocal laser-scanning microscope were deconvoluted using Leica application Suite X, 3.5.519976.

### 2.9. Cell Viability Assay

Cell viability assay was measured by alamarBlue (Thermo Fisher Scientific, Waltham, MA, USA, DAL1025). MCF7 breast cancer cells were plated in 6-well plates at 30,000 cells/well and cultured in DMEM + 10% FBS + 1× Penicillin/Streptomycin. After one week, wells were treated with either 10 μM small molecule (*n* = 2) or DMSO (*n* = 6). At the same time, 10× of alamarBlue viability reagent was added to the media to a final concentration of 1×. Fluorescence measurement was taken 1 h, 24 h, and 48 h after addition of the reagent. Fluorescence measurements were read on an EnVision Plate Reader (Perkin Elmer, Waltham, MA, USA) plate reader after transferring 50 μL of media from each condition to 96-well microplate reader. Compared to control, decreased fluorescence signal in media indicates cell death.

### 2.10. Statistical Analysis

All data are presented as the mean of *n* ≥ 3 experiments with the standard error of the mean (SEM) indicated by error bars, unless otherwise indicated. Statistical significance was determined using the chi-square test (between timepoints) or Student’s *t*-test (between conditions). Only *p* values of 0.05 or lower were considered statistically significant (*p* > 0.05 (ns, not significant) * *p* ≤ 0.05, ** *p* ≤ 0.01, *** *p* ≤ 0.001, **** *p* ≤ 0.0001). For analysis of quiescence, only fold changes of ≥1.6 were evaluated for significance. Data were compiled and analyzed with Excel for Mac (2021; Microsoft, Seattle, WA, USA).

## 3. Results

### 3.1. Small Molecule Drug Candidates Predispose GSCs to Apoptosis in The Drosophila Testis

The *Drosophila* testis contains ~10–12 GSCs in contact with the somatic hub [27]. Like other stem cells, *Drosophila* GSCs also resist drug-induced apoptosis [3,4], so we sought small molecules that can overcome that resistance and trigger apoptosis in stem cells. In this primary screen, small molecule chemicals from the NCI Diversity Set IV were mixed into low melt agar food in 96-well plates for a medium-throughput screen [25,28] (Figure 1A). Male flies were fed drug-spiked low melt food for three days, and after that, the GSCs were analyzed by staining against cleaved Dcp1 as a readout for caspase activation in apoptotic cells (Figure 1B). As previous studies have observed [29], the male GSCs showed no apoptosis in response to DMSO (Figure 1B). However, 8 out of 470 drugs tested from the NCI Diversity Set IV (Appendix A) promoted cleavage of caspase in male GSCs, while sparing the somatic hub cells (Figure 1C,D). The well-characterized topoisomerase I inhibitor, camptothecin [30], which appears in the NCI Diversity Set IV and analogues of which are now approved chemotherapeutic drugs, came out as a positive hit which potently activated apoptosis in male GSCs (Figure 1B). Additionally, seven other small molecules led to dramatic apoptosis, despite not sharing structural homology with camptothecin, suggesting unique mechanisms of action. Hence, our primary screen identified candidate drugs that activate apoptosis in stem cells, with low off-target toxicity in somatic cells, and confirms desirable drug qualities, such as oral bioavailability and metabolic stability. Further work is needed to understand how human adults tolerate these drug candidates and their derivatives. Indeed, two pairs of compounds, NSC-37168 with -37187 and NSC-98938 with -149286, have similar backbones, but with unique functional groups, which might suggest similar mechanisms of action within each pair. 

### 3.2. Small Molecule Drugs Sensitize GSCs to Insult-Induced Apoptosis in The Drosophila Ovary

Having evaluated the activity of the candidate drugs in *Drosophila* male GSCs, we next sought to characterize whether these compounds that trigger drug-induced apoptosis synergize with other pro-apoptotic stimuli like metabolic and genotoxic stresses. Like the male GSC niche, the female GSC niche is among the best characterized stem cell niches [27]. The *Drosophila* ovary is comprised of ovarioles, each of which harbors ~2–3 GSCs adjacent to the cap cells (Figure 2C and Appendix A), analogous to the somatic hub cells in the testis [27]. In this secondary screen, small molecules were mixed into grape juice, rather than low-melt agar food, therefore limiting protein and lipid intake. Flies were fed drug-spiked grape juice for two days, treated with gamma irradiation, and returned to drug-spiked juice for two more days (Figure 2A).

At 2 days post-insult (2 dpi), we found only 5% of control GSCs were apoptotic (Figure 2B). Rapamycin treatment further decreased apoptotic GSCs (1.5%), suggesting that mTORC1 inhibition can suppress or delay apoptosis in GSCs (Figure 2B). This result is consistent with the previous findings that transient mTORC1 inhibition suppresses apoptosis and promotes stem cell survival [31,32,33,34]. In contrast to rapamycin, the small molecule candidate drugs potentiated apoptosis, ranging from 11.5% apoptotic GSCs (NSC-149286) up to 30.5% (NSC-37187) (Figure 2B,C). This further suggests that the mode of action of these drugs is different than mTORC1 inhibition, perhaps through mitochondrial stress, ROS signaling or metabolic stress.

This screen further validated the drug-like activity of these small molecule compounds, now in female GSCs. Moreover, in this in vivo model, we observed synergy between the small molecules and radiation-induced apoptosis. These data further substantiate the therapeutic potential of these candidates as chemotherapeutic drugs.

### 3.3. Small Molecule Drug Candidates Inhibit Human Breast Cancer Organoid Formation

After characterizing the effect of small molecules in fruit fly stem cells, we next sought to validate these findings in a human cancer model. We tested five of our hit compounds (NSC-37168, NSC-37187, NSC-98938, NSC-149286, NSC-277184) on human cancer stem cells in vitro using MCF7 breast cancer cells (Figure 3A). 

We first showed that the small molecules at 10 μM do not affect cell viability of MCF7 cells at confluence in monolayer, as assessed by alamarBlue (Figure 3B). By contrast, Triton X-100 (0.02% or 0.1%) causes a dramatic loss of viability (Figure 3B). As MCF7 cells in monolayer are mostly comprised of non-stem cells [23,26], this finding supports our discovery in fly that these drugs preferentially sensitize rapidly dividing stem cells to apoptosis, while sparing the non-stem cells. However, we find the drugs start to show more dramatic effects when we enrich for CSCs by switching to nonadherent plates. When MCF7 cells are seeded as single cells on ultra-low attachment plates, only the stem-like cell subpopulation of MCF7 can survive in suspension and outgrow into breast cancer organoids called mammospheres (Figure 3A,C,D) [26]. After five days of undisturbed growth, the control mammospheres are large and round, and have well-defined borders (Figure 3C). On the contrary, rapamycin treatment restricted mammosphere size and led to small, underdeveloped spheres (Figure 3C).

Remarkably, treatment with the five small molecule drug candidates dramatically reduced mammosphere formation (Figure 3C,D). A subset of our small molecules—NSC-98938, NSC-149286, NSC-277184—showed strong efficacy at 1 μM concentration. Thus, it is evident that these small molecule drugs not only trigger apoptosis in adult stem cells in vivo but also impede breast cancer organoid formation in vitro. This is especially remarkable given that the same small molecules at 10 μM do not impair cell viability in MCF7 cells grown to confluence in monolayer (Figure 3B), suggesting that the identified small molecules act specifically on CSCs without nonspecific cytotoxicity in more differentiated cells. It will be interesting to explore whether protein nanocage-based drug delivery could be leveraged to direct the drug to a combination of extracellular receptors unique to the cancer or even the CSCs [35,36]. These data show that the identified small molecules act specifically on CSCs without nonspecific cytotoxicity in somatic cells.

### 3.4. Chemical-Genetic Interactions in Drosophila GSCs Suggest Unique Mechanisms between Small Molecules

To understand the mechanisms-of-action for these drugs, we next sought to characterize the effect of the small molecule drugs on stem cell quiescence. For this purpose, we used the knockdown lines where the components of critical pathways that are essential for quiescence are reduced. We tested the effect of small molecules on quiescence or apoptosis in the sensitized backgrounds with the RNAi-mediated knockdowns of Tsc1 (mTORC1 pathway component), Atg3 (autophagy component), or IKKε (NF-κB pathway activator). Since the grape juice diet proved lethal for many of our RNAi lines (data not shown), likely because of the combined metabolic stress, we fed the flies small molecules dissolved in fresh wet yeast paste instead of grape juice and dissected them at 1 dpi or 2 dpi timepoints (Figure 4A). Each GSC contains a spectrosome, which is usually rounded but elongates during cell division (Figure 2C and Appendix A) [5,6,21,22]. Using spectrosome morphology as an indicator, unirradiated GSCs divide regularly (~26%), arrest division at 1-day post-insult (~6%), and resume dividing at 2-days post-insult (~26%) (Figure 4G), the hallmark of functioning GSC quiescence after insult [5,6]

The knockdown lines of Tsc1 or Atg3 fed with DMSO showed similar levels of GSCs division between 1 dpi and 2 dpi, indicating a lack of quiescence despite the irradiation (Figure 4B,C). This agrees with previous findings that mTORC1 repressors and autophagy genes are required for GSC quiescence [26]. Unsurprisingly, when the flies were fed with rapamycin-spiked yeast, Tsc1 KD GSCs can now enter quiescence at 1 dpi, further confirming that Tsc1-mediated mTORC1 inhibition is essential for insult-induced quiescence (Figure 4B). Interestingly, rapamycin also reduced division at 1 dpi in Atg3 KD (Figure 4C), suggesting that autophagy promotes quiescence in a mTORC1-dependent manner. However, our small molecule drugs (NSC-37168, NSC-37187, NSC-127458, NSC-277184) failed to restore quiescence in either Tsc1 KD or Atg3 KD (Figure 4B–E). Notably, NSC-277184 showed increased division at 2 dpi in Tsc1 KD GSCs, while not affecting the division at 2 dpi in Atg3 KD GSCs (Figure 4D,E). Further, these data suggest that NSC-277184 might mechanistically inhibit autophagy which leads to an increase in the cell division in Tsc1 KD GSCs whereas NSC-277184 fails to further inhibit autophagy in Atg3 RNAi hence does not show further changes in division in Atg3 KD GSCs (Figure 4B,C).

Since the identified small molecules did not show interactions with previously identified genetic pathways in the process, we screened for additional alternative pathways that might be crucial for quiescence. Past work has demonstrated that loki/dmChk2, a DNA damage-responsive kinase, is essential for GSCs quiescence after irradiation [5,22]. Accordingly, we decided to probe the function of a different DNA damage repair protein, Ogg1, a tumor suppressor and glycosylase involved in the repair of ROS-induced 8-oxoG base lesions [37,38,39]. In contrast to control, when we knocked down Ogg1 by RNAi, GSC division remains high at 1 dpi (~16%), suggesting that Ogg1-mediated base excision repair is required for GSC quiescence but without a decrease in GSCs number (Appendix A). Next, studies also suggest that cells enter quiescence with a concomitant change in cytoplasmic pH [40]. Therefore, we were interested to test if pH as a factor can affect GSCs entry into quiescence upon irradiation, so we tested Nhe3 (Na+/H+ exchanger). In contrast to control, we found that knockdown of the Nhe3 [41] also leads to sustained division at 1 dpi (~20%), but GSC division drops lower (~12%) at 2 dpi, suggesting these GSCs take a long time to enter quiescence (Appendix A). Though Ogg1 KD and Nhe3 KD abolished GSC quiescence, the GSC number remained unchanged before and after insult, similar to control (Appendix A).

Next, we identified that the nuclear factor-κB (NF-κB) pathway is critical for GSC quiescence, as knockdown of non-canonical activator, ΙΚΚε [42], results in failure to arrest GSC division at 1 dpi (~20%) (Figure 4G). Remarkably, knockdown of ΙΚΚε also resulted in increased apoptosis, as inferred from the GSC number dropping from before irradiation (~2.1 GSC/germarium) to 2 dpi (~1.3 GSC/germarium) (Figure 4H). This finding suggests that ΙΚΚε and downstream NF-κB activation help GSCs to enter quiescence and protect them from apoptosis; this is consistent with previous reports that ΙΚΚε acts as a pro-survival oncogene in triple negative breast cancer [43]. Further work is required to understand whether other NF-κB pathways components are similarly anti-apoptotic after insult in GSCs. Our observation further suggested that a down-regulation of Ogg1-mediated DNA repair or Nhe3 mediated pH changes did not induce apoptosis. In contrast, down-regulation of NFκB pathway induces apoptosis in GSCs. Therefore, we utilized the sensitized ΙΚΚε KD GSCs background and tested if our small molecule drugs could synergize with NF-κB-mediated apoptosis and further activate cell death. To this end, we fed ΙΚΚε KD flies with small molecule drugs (NSC-37168, NSC-37187, NSC-127458, NSC-277184), or rapamycin. We found that rapamycin treatment also restores the quiescence in ΙΚΚε KD upon 2 dpi, suggesting that NF-κB pathway is upstream of mTORC1 in the maintenance of quiescence (Figure 4I). However, we did not observe any change in GSCs division for any other drugs used (Figure 4I,K). Interestingly, we observed a decrease in the GSCs number in the NSC-37168 fed flies, suggesting that NSC-37168 promotes loss of GSCs in ΙΚΚε KD (Figure 4J,K). To test if the loss of GSC was caused by differentiation or by GSC cell death, we analyzed cDCP levels in the GSC and observed that the levels of cDCP are enhanced in the flies fed with the NSC-37168 (Appendix A). These data suggest that NSC-37168 cross-talks with the NF-κB pathway downstream of ΙΚΚε to promote apoptosis in the GSCs.

## 4. Discussion

Cancer-initiating cells/CSC are a subpopulation of cancer cells with the unique ability to avoid apoptosis, produce diverse progeny, and metastasize. The plasticity and heterogeneity of these cells support tumor progression, drug resistance, and cancer recurrence; therefore, there’s a crucial need for CSC-specific therapies. Our study took a novel approach to discover chemotherapy drug candidates, and druggable pathways that sensitize CSCs to apoptosis.

In this work, we report novel bioactivities of several small molecule drug candidates with high therapeutic potential for cancer treatment. Notably, we have performed a three-level screen to identify and characterize the candidate small molecule drugs using *Drosophila* GSCs and human breast cancer cell organoids. We have also identified a subset of small molecules affecting CSC viability (NSC-94600, NSC-37168, NSC-37187, NSC-98938, NSC-125197, NSC-149286, NSC-277184) (Appendix A). Furthermore, we have established *Drosophila* GSCs as an in vivo model to screen for drugs targeting cancer stem cells. In addition, our data also confirm that mTOR activity is pivotal for stem cells to enter quiescence. Interestingly, we have found that the knockdown of Atg3 or IKKε which are essential for quiescence can be rescued by mTOR inhibition. This suggests that autophagy and NF-κB pathways are upstream of mTOR in the maintenance of quiescence and prevent apoptosis. Thus, our data reveal the crosstalk between various signaling pathways in the maintenance of stem cells.

Frequently, CSCs undergo cellular reprograming and co-opt existing stem cell regulatory pathways like TGF-β, Wnt/β-catenin, TNFα, and NF-κB. Using various gene knockdowns, including Tsc1, Atg3, or IKKε, we have further teased apart the cellular pathways and how they can be targeted by drugs. Our data suggest that NSC-277184 likely acts through autophagy as we have observed an increase in GSCs division in Tsc1 KD and IKKε KD but not in Atg3 KD. Further studies are required to understand whether other small molecules identified in this study interact with specific pathways.

Strikingly, we show that NF-κB pathway is critical for GSC quiescence and that the small molecule, NSC-37168 cross-talks with NF-κB pathway. Importantly, ΙΚΚε KD already shows a decrease in GSCs number, but the drug NSC-37168 treatment further significantly reduces this number (Figure 4H,J). indicating that NSC-37168 enhances apoptosis in ΙΚΚε KD flies. Importantly, this effect was specific for NF-κB pathway since NSC-37168 did not induce further cell death in Atg3 KD or Tsc1KD backgrounds. This not only reveals that NF-κB pathway is critical in GSCs, and plausibly CSCs, but also indicates that this fundamental stemness pathway is targetable by the identified drugs. Moreover, NF-κB has been known to transcriptionally activate genes like CD44, MMP9, and Atg3 that contribute to the stemness of CSCs (Figure 4F) [44,45,46]. NF-κB is known to upregulate growth factor receptor, CD44 expression which is a membrane marker for CSCs [44,46]. Furthermore, NF-κB also upregulates autophagy through Atg3, and we have also established and characterized Atg3/autophagy as a key regulator of GSC quiescence [6]. Further, it will be interesting to test the functional importance of other NF-κB target genes in GSC quiescence. Additionally, NF-κB also regulates extracellular matrix through MMP9 and promote tumor invasion [45]. We propose that NSC-37168 crosstalks with NF-κB pathway and might also affect the tumorigenic transcriptional activity of NF-κB. Further studies are needed to understand the precise molecular impact of the NSC-37168 drug. Our study underscores the value of the *Drosophila* model for uncovering novel chemical-genetic interactions and discovering viable drug candidates. We propose that these drugs have the potential to prevent metastasis and tumor relapse if used in the proper combination therapy.

## Figures and Tables

**Figure 1 cells-10-02771-f001:**
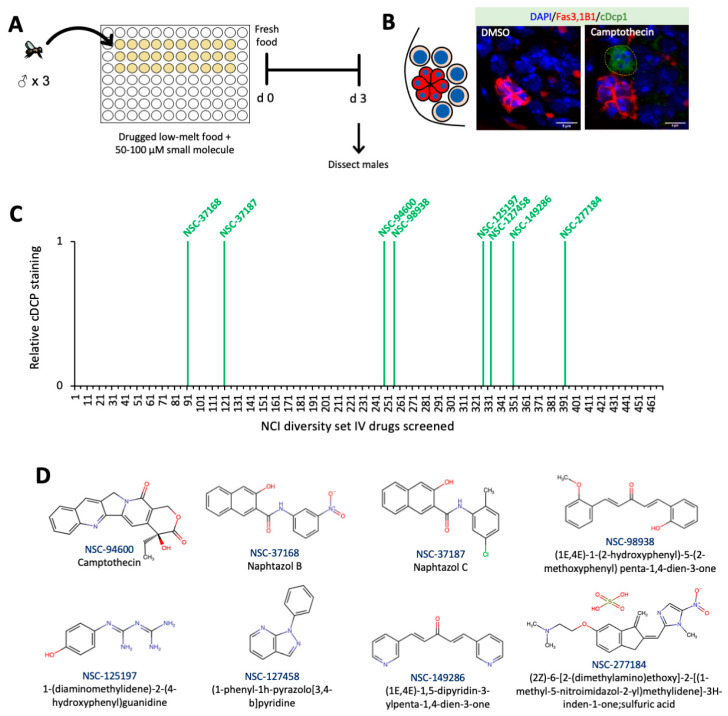
Primary screen for GSC apoptosis in male fruit flies fed drug-spiked solid food (**A**) Diagram of experimental set up for NCI Diversity Set IV small molecule screen in low melt agarose food in male GSCs. (**B**) Diagram of male *Drosophila* GSC niche, wherein 6–12 GSCs (beige) are adjacent to niche cells (red), with all cells containing nuclei (blue). Representative images of germline from males treated with either vehicle control (DMSO) or camptothecin/NSC-94600. Stained with 1B1 (spectrosome, red), FasIII (hub, red), cDcp1 (apoptosis, green), and DAPI (nuclei, blue) (Scale bar 5 μm). (**C**) Visual representation of small molecules from the NCI Diversity Set IV which cause Dcp1 cleavage in male GSCs. (**D**) Chemical structures of small molecules that potentiated apoptosis in the male GSCs in the primary drug screen.

**Figure 2 cells-10-02771-f002:**
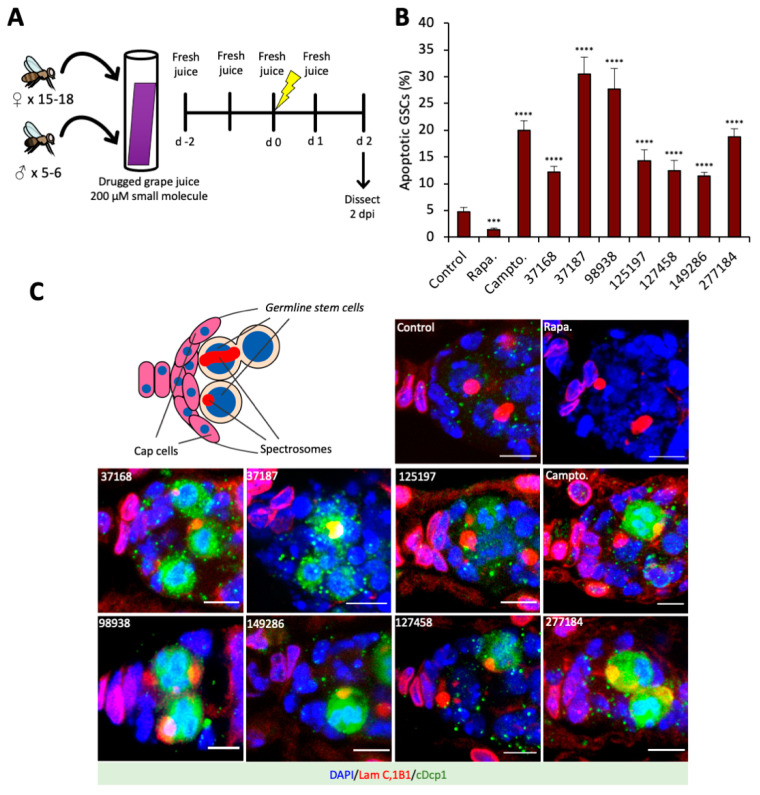
Secondary screen for GSC apoptosis in irradiated female fruit flies fed drug-spiked grape juice (**A**) Diagram of experimental set up for small molecule screen of pro-apoptotic drugs in grape juice in female GSCs. New grape juice containing 200 μM of respective drugs is given to flies every day during the irradiation treatment. Flies are dissected for ovaries at 2 dpi. (**B**) Percentage of apoptotic GSCs, computed as a ratio of cDcp+ positive GSCs to all GSCs. (**C**) Cartoon model of the GSC niche depicting spectrosome-containing GSCs in contact with somatic cap cells, and representative images of germaria with apoptotic female GSCs after specified drug treatment and post-irradiation (2 dpi). Stained with 1B1 (spectrosomes/fusomes, red), LamC (Cpc and TF, red), cDcp1 (apoptosis, green) and DAPI (nuclei, blue). (Scale bar 5 μm). Significance is calculated by Chi-squared test between control (DMSO) and each drug treated condition, *** *p* ≤ 0.001, **** *p* ≤ 0.0001.

**Figure 3 cells-10-02771-f003:**
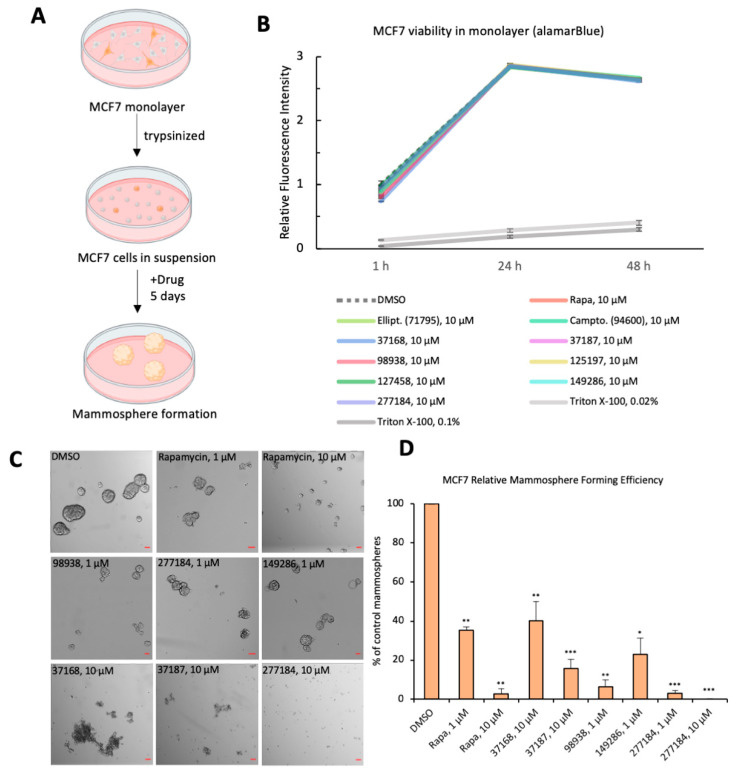
Tertiary screen for CSC impairment in MCF7 breast cancer organoids (**A**) Schematic representation of the experimental procedure for MCF7 organoid formation and candidate drug treatment for various concentrations. (**B**) alamarBlue cell viability curve for MCF7 cells in monolayer treated with either vehicle (DMSO), small molecules, or Triton X-100. Relative fluorescence intensity was computed as ((absorbance[condition] − absorbance[blank])/(absorbance[DMSO, 1 h]). (**C**) Brightfield images of MCF7-derived mammospheres after being treated with the indicated each small molecule drug candidates or vehicle (DMSO) and their specified concentration (Scale bar 100 μm). (**D**) Quantification of mammospheres >250 μm in diameter. Relative mammosphere formation efficiency (MFE) is computed as a percentage of control (no. of mammospheres [condition]/no. of mammospheres [DMSO] ∗ 100%). Significance is calculated by Student’s *t*-test between control (DMSO) and each drug treated condition), * *p* ≤ 0.05, ** *p* ≤ 0.01, *** *p* ≤ 0.001.

**Figure 4 cells-10-02771-f004:**
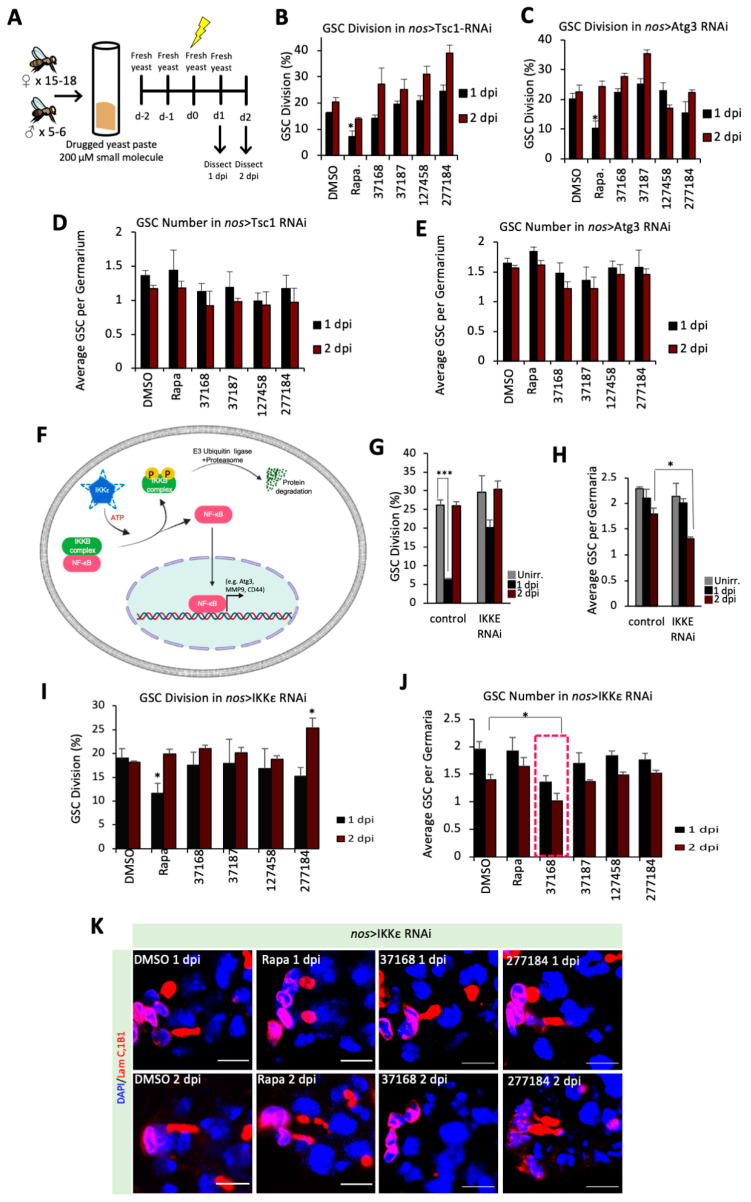
Pathway analysis in RNAi gene knockdown GSCs in irradiated females fed drug-spiked yeast (**A**) Diagram of experimental set up of yeast paste drug treatment. New yeast paste containing 200 μM of respective drugs or vehicle (DMSO) is given to flies every day during the irradiation treatment. Flies are dissected for ovaries at 1 dpi and 2 dpi. (**B**) Percentage of GSC division at 1 dpi and 2 dpi in Tsc1 RNAi knockdown, treated with vehicle (DMSO), rapamycin (200 μM), or one of the candidate drugs (200 μM). (**C**) Percentage of GSC division rate in Atg3 RNAi, treated with vehicle (DMSO), rapamycin (200 μM), or one of the candidate drugs (200 μM). (**D**) GSC number in Tsc1 RNAi knockdown, treated with vehicle (DMSO), rapamycin (200 μM), or one of the candidate drugs (200 μM). (**E**) GSC number in Atg3 RNAi, treated with vehicle (DMSO), rapamycin (200 μM), or one of the candidate drugs (200 μM). (**F**) Schematic representation of NF-κB pathway regulation by IKKε: IκB complex sequesters NF-κB to the cytoplasm. IKKε has been shown to phosphorylate the IκB complex, targeting it for degradation, and allowing NF-κB to translocate to the nucleus and activate target genes. (**G**) Percentage of GSC division rate in control versus IKKε RNAi GSCs when unirradiated, at 1 dpi, and at 2 dpi, significance calculated in control between Unirr. and 1 dpi by Chi-squared test, *** *p* ≤ 0.001.(**H**) Average GSC number in control versus IKKε RNAi. (**I**) Percentage of GSC division rate in IKKε RNAi, treated with vehicle (DMSO), rapamycin (200 μM), or one of the candidate drugs (200 μM). (**J**) Average GSC number in IKKε RNAi knockdown, treated with vehicle (DMSO), rapamycin (200 μM), or one of the candidate drugs (200 μM). GSC number is reduced upon co-treatment with NSC-37168 (red dashed box). (**K**) Representative confocal image of germarium stained for 1B1 (spectrosomes/fusomes, red), LamC (Cpc and TF, red) and DAPI (nuclei, blue) upon various drug treatment (200 μM) at 1 dpi or 2 dpi. (Scale bar 5 μm). Significance is calculated by Chi-squared test between control (DMSO) and each drug treated condition, * *p* ≤ 0.05, *** *p* ≤ 0.001.

## Data Availability

Data is contained within the article or Appendix A. The data presented in this study are available in Appendix A.

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
