# Peer review of "Chemical Genetic Screen in Drosophila Germline Uncovers Small Molecule Drugs That Sensitize Stem Cells to Insult-Induced Apoptosis"

_cells, 2021, doi:10.3390/cells10102771_

Round 1
Reviewer 1 Report
In this paper, Ishibashi JR and colleagues performed a chemical/genetic screen to identify drugs and signaling pathways required to sensitize adult stem cells to apoptosis. To do so, they combined various and complementary approaches. They first set a primary screen using the Drosophila male GSC (from testis) as a model, which is known to resist drug induced apoptosis. They managed to pinpoint 8 small molecule candidates that trigger GSC apoptosis. Building up on this screen, they subsequently turned to the females GSC system, a suitable assay to investigate whether there is any synergy between the drug induced apoptosis and genotoxic stress. They found that the previously identified molecules can synergize with the metabolic and genotoxic stresses to induce the apoptosis GSC. Importantly, they validated their candidates in a cell culture model of breast cancer organoid formation. Finally, they showed that NF-kB is an important pathway in GSC quiescent and viability.
In summary, this is a very interesting and well-designed study that may have wider implications for understanding how other adult stem cells and cancer stem cells may resist apoptosis. The data are good quality and well-presented. I list here some minor suggestions; I hope they can be useful to consolidate the manuscript.
Figure 1. The authors should clarify a) whether the identified small molecules influence the animal viability/longevity? b) to which extent the somatic cells enter apoptosis too?
Figure 2. It is recommended to have an additional control where the flies are exposed to drugs without being radiated. The prediction would be that the percentage of apoptotic cells will be slightly lower comparing to drugs + gamma radiations (Fig2 B). This experiment should consolidate the conclusion that there is a synergy between the drugs intakes and genotoxic stresses.
Figure 4. For the sake of clarity, the authors should better explain, in the main text, how the dividing GSC were visualized (PH3? Edu?), and if possible, provide a representative example (image) of the experiment.
Minor comments.
Figure 2. To help the non-expert readers in understanding the system, I suggest to add an ovariole cartoon showing the cap cells/GSC. As shown for the males in Fig 1B.
Figure 3. Please describe what are the green cells in Fig 3?
Figure 4. Please standardize the Y axis titles in Fig 4B vs 4C.
Line 276. The reference 34 is not well formatted.
Author Response
Reviewer #1
In this paper, Ishibashi JR and colleagues performed a chemical/genetic screen to identify drugs and signaling pathways required to sensitize adult stem cells to apoptosis. To do so, they combined various and complementary approaches. They first set a primary screen using the Drosophila male GSC (from testis) as a model, which is known to resist drug induced apoptosis. They managed to pinpoint 8 small molecule candidates that trigger GSC apoptosis. Building up on this screen, they subsequently turned to the females GSC system, a suitable assay to investigate whether there is any synergy between the drug induced apoptosis and genotoxic stress. They found that the previously identified molecules can synergize with the metabolic and genotoxic stresses to induce the apoptosis GSC. Importantly, they validated their candidates in a cell culture model of breast cancer organoid formation. Finally, they showed that NF-kB is an important pathway in GSC quiescent and viability.
In summary, this is a very interesting and well-designed study that may have wider implications for understanding how other adult stem cells and cancer stem cells may resist apoptosis. The data are good quality and well-presented. I list here some minor suggestions; I hope they can be useful to consolidate the manuscript.
Figure 1. The authors should clarify a) whether the identified small molecules influence the animal viability/longevity?
Response: We thank the reviewer for this question. We have tested the viability of the animals and show that these small molecules do not significantly influence animal viability. We have now included a graph (Fig-S1B) showing the effect of small molecules on organismal viability.
- b) to which extent the somatic cells enter apoptosis too?
Response: The goal of the primary screen was to identify the small molecules that did not show significant defects in the differentiated cells, but affected the GSC viability.
Figure 2. It is recommended to have an additional control where the flies are exposed to drugs without being radiated. The prediction would be that the percentage of apoptotic cells will be slightly lower comparing to drugs + gamma radiations (Fig2 B). This experiment should consolidate the conclusion that there is a synergy between the drugs intakes and genotoxic stresses.
Response: We have now performed the experiment suggested by the reviewer and we observe higher level of death with IR+drug than just drug alone, as predicted by the reviewer.
Figure 4. For the sake of clarity, the authors should better explain, in the main text, how the dividing GSC were visualized (PH3? Edu?), and if possible, provide a representative example (image) of the experiment.
Response: We have now clarified that spectrosome morphology (rounded vs. elongated) is our criterion for determining cell division (Fig. 2C, Fig 3A)
Minor comments.
Figure 2. To help the non-expert readers in understanding the system, I suggest to add an ovariole cartoon showing the cap cells/GSC. As shown for the males in Fig 1B.
Response: As suggested by the reviewer, we have now added a model in Fig 2C depicting the cap cells, GSCs, and other important cells in a female germarium.
Figure 3. Please describe what are the green cells in Fig 3?
Response: We have now more clearly described the cells in Fig.3. The minority of orange-colored MCF7 cells can initiate mammospheres, while the majority of green MCF7 cells can’t initiate mammospheres.
Figure 4. Please standardize the Y axis titles in Fig 4B vs 4C.
Response: We have now standardized the Y axis in Fig. 4B and 4C.
Line 276. The reference 34 is not well formatted.
Response: We have now formatted the reference 34 better.

Reviewer 2 Report
The manuscript presented by Ishibashi et. al., describes a novel genetic screen to identify drugs that induce germline stem cells, normally refractory, to undergo apoptosis. The drug administration regime and screening process will be quite useful to other investigators. The authors then go on to validate the selected drugs in MCF-7 organoids. I appreciate the possibilities of conducting the screen in a wild type stem cell niche afforded by the Drosophila experiments but if the ultimate aim was to identify drugs that affects organoid formation then why not conduct the screen in vitro? Perhaps a little more reframing of the narrative would be useful to present the novelty of conducting these analyses of stem cells, within their niche, where they are regulated by all of the normal endogenous factors.
The results are generally clearly presented and demonstrate that eight compounds can induce germline stem cells to initiate apoptosis, as measured by cleaved Dcp-1 staining. I am unsure of what the graph in Fig 1C is showing – is this simply Yes/No for Dcp-1? If this is the case I think it would be better presented as a table with +/- indications. Figure legend 1 mentions germaria when referring to testes. This is a female-specific term.
The Figure 2 data are very nicely presented and definitively show the relative increase in GSC apoptosis associated with each drug.
Panel B in Figure 3 indicates that MCF7 viability was identical under numerous conditions (I presume that the lines on the plot overlay one another). Could this be plotted differently to indicate the data points? It is is not clearly explained in Panel 3D whether the number of mammospheres or the size of mammospheres are being quantified. I presume it to be the former, as less CSC activity would seed fewer mammospheres. Could the drugs be affecting the proliferation of progeny cells produced by the CSCs and therefore leading to fewer mammospheres >250uM in diameter, or is size also a reflection of CSC activity? Perhaps a little more explanation may help to justify the statement “These data show that the identified small molecules act specifically on CSCs without nonspecific cytotoxicity in somatic cells”
The statement “In contrast to control, we found that knockdown of the Nhe3 [20] also leads to sustained division at 1dpi (~20%), but GSC division drops lower (~12%) at 2dpi, suggesting these GSCs take a long time to enter quiescence or possibly differentiate prematurely (Fig. S1C,D)” suggests that the drop at 2dpi is significant – is this the case?
Overall this study is quite interesting and could benefit from minor improvements suggested above.
Author Response
Reviewer #2
The manuscript presented by Ishibashi et. al., describes a novel genetic screen to identify drugs that induce germline stem cells, normally refractory, to undergo apoptosis. The drug administration regime and screening process will be quite useful to other investigators. The authors then go on to validate the selected drugs in MCF-7 organoids. I appreciate the possibilities of conducting the screen in a wild type stem cell niche afforded by the Drosophila experiments but if the ultimate aim was to identify drugs that affects organoid formation then why not conduct the screen in vitro?
Response: As the reviewer appreciates, studying the effects of drugs in the model organism is important and more physiological than cell culture since it allows us to study the stem cells in their natural niche, surrounded by endogenous growth factors and signals.
Perhaps a little more reframing of the narrative would be useful to present the novelty of conducting these analyses of stem cells, within their niche, where they are regulated by all of the normal endogenous factors.
Response: We have now reframed the narrative in the introduction.
The results are generally clearly presented and demonstrate that eight compounds can induce germline stem cells to initiate apoptosis, as measured by cleaved Dcp-1 staining. I am unsure of what the graph in Fig 1C is showing – is this simply Yes/No for Dcp-1? If this is the case I think it would be better presented as a table with +/- indications. Figure legend 1 mentions germaria when referring to testes. This is a female-specific term.
Response: We thank the reviewer pointing this out. We have now made the corrections. Fig-1C is binary for cDcp1 signal. As suggested by the reviewer, we have now also included a supplemental table (S.Table1) listing all the drugs with +/- indication for cDcp1.
The Figure 2 data are very nicely presented and definitively show the relative increase in GSC apoptosis associated with each drug. Panel B in Figure 3 indicates that MCF7 viability was identical under numerous conditions (I presume that the lines on the plot overlay one another). Could this be plotted differently to indicate the data points? It is is not clearly explained in Panel 3D whether the number of mammospheres or the size of mammospheres are being quantified.
Response: We appreciate reviewer’s comments on Fig.2. To clarify Fig.3B,D we have now described the quantification of the mammosphere # in Fig.3D in more detail and report the primary data for monolayer viability (Fig.3B) in S.Table2.
I presume it to be the former, as less CSC activity would seed fewer mammospheres. Could the drugs be affecting the proliferation of progeny cells produced by the CSCs and therefore leading to fewer mammospheres >250uM in diameter, or is size also a reflection of CSC activity? Perhaps a little more explanation may help to justify the statement “These data show that the identified small molecules act specifically on CSCs without nonspecific cytotoxicity in somatic cells”
Response: We have now added further explanation as suggested by the reviewer. The reviewer is correct, we cannot distinguish by our assay whether the drug kills the CSC or drives it to quiescence since we only report the number of large spheres (>250um). Rapamycin treatment, we believe mainly affects the size of the mammospheres, suggesting that its mode of action is cell cycle dependent in this assay.
The statement “In contrast to control, we found that knockdown of the Nhe3 [20] also leads to sustained division at 1dpi (~20%), but GSC division drops lower (~12%) at 2dpi, suggesting these GSCs take a long time to enter quiescence or possibly differentiate prematurely (Fig. S1C,D)” suggests that the drop at 2dpi is significant – is this the case?
Response: Yes, the drop at 2dpi is significant. We have now added this information in the figure.
Overall this study is quite interesting and could benefit from minor improvements suggested above.
Response: We thank the reviewer for these encouraging comments.

Reviewer 3 Report
In their manuscript entitled “Chemical genetic screen in Drosophila germline uncovers small molecule drugs that sensitize stem cells to insult-induced apoptosis,” Ishibashi, Keshri, Taslim et al. present a 3-part screen of drug candidates that may promote cancer stem-cell apoptosis when combined with a secondary insult such as irradiation. They identify 7 new candidate drugs, confirming in vivo function using Drosophila and potential functionality in the human system using a breast-cancer cell line. They further attempt to identify possible mechanism-of-action for the drugs, and identify the NF-kB pathway to play a role in GSC quiescence.
The manuscript overall is a nice example of the power and potential of the Drosophila system for screening drug candidates that are likely to have in vivo efficacy and have a potentially better chance of passing through clinical trials than drugs identified in vitro or in silico. The identified candidates will be of interest to the cancer field and will likely lead to further studies on suitability as chemotherapy agents as well as to better characterize mechanism of action. In addition, the study provides further insight into the signaling pathways regulating quiescence in the Drosophila GSC, and may be of interest to the Drosophila stem cell community. However, I feel there are several issues that need to be addressed, in particular with providing sufficient background.
- The introduction should be expanded to provide a better overview of what is known in the field regarding NF-kB and mTOR signaling and their regulation of GSC/CSC proliferation and apoptosis. There should also be an expanded introduction of the Drosophila GSC as a model to study quiescence and apoptosis and screen small molecules.
- Line 226: The tested drugs have the opposite effect on apoptosis as rapamycin, which inhibits mTORC1, but this doesn’t mean that they are antagonistic to mTORC. This experiment does not address the relationship between mTORC signaling and the drug candidates.
- Line 229: Synergy would imply the effect of drug + irradiation is much greater than either irradiation or drug alone. These data are not provided, so it is not possible to judge if treatment with the small molecules alone would illicit the same level of apoptosis without irradiation.
- Figure 3B – I can only see a line for DMSO in the plot, but it appears there are some points much lower on the plot. Does this reflect lower metabolic rates in the small molecule treated MCF7 cells? Perhaps there is a different way to visualize or color all of the different small molecule treatments and Triton X controls?
- Rapamycin treatment prevents apoptosis in the GSC, but also prevents expansion of MCF7 cells to form mammospheres. By contrast, the small molecules both impede mammosphere formation and promote apoptosis in the GSC. Can the authors provide some explanation for this difference?
- Line 260-261: The authors suggest the small molecules tested are not cytotoxic in somatic cells. More generally though, what is the logic or subsequent approach to tailor these molecules to be CSC specific? I assume there would be severe side effects if existing stem cells in the human body were pushed to undergo apoptosis, in addition to the CSC population.
- Line 272-274 – There should be a sentence added here to explain more directly that the GSC in irradiated females at 1 dpi enters quiescence, and after damage is repaired, increases division again at 2 dpi. This is currently first explained in line 294-295.
- Figure 4 – Please clearly mark which bars and which comparisons are significant (1 dpi vs 2 dpi or DMSO control vs drug?). There are several places in the text where the authors reference a result from these plots that is not marked as significant. For example, Lines 285-287: the data on NSC-277184 is not marked as significant, yet the authors discuss increased cell division in the Tsc1 KD GSCs and conclude a different to Atg3 RNAi knockdown. This is also the case for Lines 374-376 in the discussion. For Figure 4B-E, the authors should provide the driver only, Tsc1-RNAi only and Atg3 RNAi only controls, to show that they do enter quiescence normally.
- The section of the results from line 273-306 could be split into different paragraphs and modified for better clarity. Was this part of a larger screen for multiple signaling pathways, or was this a candidate approach with only the three components mentioned in the text? How are the Ogg1 and Nhe3 data related to the small molecule screening? The logic of including this data needs to be explained more clearly. Does knockdown of Nhe3 actually change the pH in GSCs?
- The claim that NSC-37168 acts through the NF-kB pathway to induce apoptosis is not well presented or supported, and seems based on a single result in the supplement. This data is also mentioned in the discussion (line 378-387) for a major conclusion. In line 380 the authors claim their data shows a role for NF-kB in CSCs, but it is unclear which data support this claim. The authors should also more clearly discuss what is already known about NF-kB in this context (both in CSCs as well as GSCs).
Author Response
Reviewer #3
In their manuscript entitled “Chemical genetic screen in Drosophila germline uncovers small molecule drugs that sensitize stem cells to insult-induced apoptosis,” Ishibashi, Keshri, Taslim et al. present a 3-part screen of drug candidates that may promote cancer stem-cell apoptosis when combined with a secondary insult such as irradiation. They identify 7 new candidate drugs, confirming in vivo function using Drosophila and potential functionality in the human system using a breast-cancer cell line. They further attempt to identify possible mechanism-of-action for the drugs, and identify the NF-kB pathway to play a role in GSC quiescence.
The manuscript overall is a nice example of the power and potential of the Drosophila system for screening drug candidates that are likely to have in vivo efficacy and have a potentially better chance of passing through clinical trials than drugs identified in vitro or in silico. The identified candidates will be of interest to the cancer field and will likely lead to further studies on suitability as chemotherapy agents as well as to better characterize mechanism of action. In addition, the study provides further insight into the signaling pathways regulating quiescence in the Drosophila GSC, and may be of interest to the Drosophila stem cell community. However, I feel there are several issues that need to be addressed, in particular with providing sufficient background.
- The introduction should be expanded to provide a better overview of what is known in the field regarding NF-kB and mTOR signaling and their regulation of GSC/CSC proliferation and apoptosis. There should also be an expanded introduction of the Drosophila GSC as a model to study quiescence and apoptosis and screen small molecules.
Response: we have now expanded the introduction as the reviewer suggests
- Line 226: The tested drugs have the opposite effect on apoptosis as rapamycin, which inhibits mTORC1, but this doesn’t mean that they are antagonistic to mTORC. This experiment does not address the relationship between mTORC signaling and the drug candidates.
Response: We have now made the corrections and reframed the sentence.
- Line 229: Synergy would imply the effect of drug + irradiation is much greater than either irradiation or drug alone. These data are not provided, so it is not possible to judge if treatment with the small molecules alone would illicit the same level of apoptosis without irradiation.
Response: We have now performed the experiment suggested by the reviewer and we observe higher level of death with IR+drug than just drug alone, as predicted by the reviewer.
- Figure 3B – I can only see a line for DMSO in the plot, but it appears there are some points much lower on the plot. Does this reflect lower metabolic rates in the small molecule treated MCF7 cells? Perhaps there is a different way to visualize or color all of the different small molecule treatments and Triton X controls?
Response: We apologize the mistake in uploading the figure. We have now re-uploaded the image and made sure all lines are visible. In addition, we have added the data points to a supplemental Table.
- Rapamycin treatment prevents apoptosis in the GSC, but also prevents expansion of MCF7 cells to form mammospheres. By contrast, the small molecules both impede mammosphere formation and promote apoptosis in the GSC. Can the authors provide some explanation for this difference?
Response: We thank the reviewer for this question. We agree that Rapamycin and the identified small molecules probably act through different modes of action. Rapamycin inhibits mTORC1 and therefore reduces cell division. In insulted GSCs, reduced cell division would both lower the severity of DNA damage at the time of insult and facilitate quiescence after the insult. However, in mammospheres, reduced cell division would restrict sphere growth, resulting in small, round spheres that don’t reach the minimum diameter, which is what we observe in Fig. 3C. In contrast, the small molecule drugs show no evidence of reducing cell division, and instead promote apoptosis in GSCs. Moreover, the structures in small molecule-treated conditions are not spherical and have rough edges and other malformations, suggesting the small molecules impair CSC viability beyond simple antiproliferative activity. Though we can’t rule out the possibility that these small molecules also reduce cell proliferation in MCF7 cells, we think it’s likely that they primarily stimulate apoptosis, like they do in the insulted GSCs.
- Line 260-261: The authors suggest the small molecules tested are not cytotoxic in somatic cells. More generally though, what is the logic or subsequent approach to tailor these molecules to be CSC specific? I assume there would be severe side effects if existing stem cells in the human body were pushed to undergo apoptosis, in addition to the CSC population.
Response: We thank the reviewer for raising this important point. We have shown that these drugs seem to preferentially sensitize stem cells, while largely sparing more differentiated cells, in both GSCs and MCF7 cells. We agree with the reviewer however that future work needs to focus on precise delivery so that healthy adult stem cells are not affected/harmed by these drugs. We have worked in collaboration with IPD (UW, Seattle) on self-assembling nanocages, which in the future can be developed to target specific cell types (CSC). These IPD protein nanocages could be leveraged for targeted drug delivery as is briefly suggested in this paper and this paper. Other laboratories are also working towards localized drug delivery using other approaches.
- Line 272-274 – There should be a sentence added here to explain more directly that the GSC in irradiated females at 1 dpi enters quiescence, and after damage is repaired, increases division again at 2 dpi. This is currently first explained in line 294-295.
Response: We have now made the suggested change.
- Figure 4 – Please clearly mark which bars and which comparisons are significant (1 dpi vs 2 dpi or DMSO control vs drug?). There are several places in the text where the authors reference a result from these plots that is not marked as significant. For example, Lines 285-287: the data on NSC-277184 is not marked as significant, yet the authors discuss increased cell division in the Tsc1 KD GSCs and conclude a different to Atg3 RNAi knockdown. This is also the case for Lines 374-376 in the discussion. For Figure 4B-E, the authors should provide the driver only, Tsc1-RNAi only and Atg3 RNAi only controls, to show that they do enter quiescence normally.
Response: We have now made these corrections.
- The section of the results from line 273-306 could be split into different paragraphs and modified for better clarity. Was this part of a larger screen for multiple signaling pathways, or was this a candidate approach with only the three components mentioned in the text? How are the Ogg1 and Nhe3 data related to the small molecule screening? The logic of including this data needs to be explained more clearly. Does knockdown of Nhe3 actually change the pH in GSCs?
Response: We have now explained the logic more clearly in the text. Nhe3 is a Na+/H+ exchanger and has been characterized for causing acidification in presence of amiloride (Nhe inhibitor) using pH sensitive fluorophore in Drosophila during oogenesis ( Wei et al., BMC Dev Biol)
- The claim that NSC-37168 acts through the NF-kB pathway to induce apoptosis is not well presented or supported, and seems based on a single result in the supplement. This data is also mentioned in the discussion (line 378-387) for a major conclusion. In line 380 the authors claim their data shows a role for NF-kB in CSCs, but it is unclear which data support this claim. The authors should also more clearly discuss what is already known about NF-kB in this context (both in CSCs as well as GSCs).
Response: We have now corrected this section. We have referenced the correct papers, where needed and expanded the description of what is already known.

Round 2
Reviewer 3 Report
The authors have addressed nearly all of my comments in their revision. The new supplemental figure 1 and the table of tested drugs, as well as the many textual revisions and additional figure revisions, have greatly improved the manuscript.
There is one comment I was unsure of, however. Regarding Figure 2, my previous comment about Line 229 with the wording of "synergy" of the drug + irradiation, the authors respond : "We have now performed the experiment suggested by the reviewer and we observe higher level of death with IR+drug than just drug alone, as predicted by the reviewer." I am unsure where this data is included in the manuscript, if it is included. If not, the authors should revise that statement in the text.